



# The effect of different Climate and Air Quality policies in China on in situ Ozone production in Beijing

Beth S. Nelson[1], Zhenze Liu[2a], Freya A. Squires[1b], Marvin Shaw[1,3], James R. Hopkins[1,3], Jacqueline F. Hamilton[1,3], Andrew R. Rickard[1,3], Alastair C. Lewis[1,3], Zongbo Shi[4], James D. Lee[1,3]

[1]Wolfson Atmospheric Chemistry Laboratories, Department of Chemistry, University of York, Heslington, York, YO10 5DD, UK.
[2]School of Geosciences, The University of Edinburgh, Edinburgh, UK.
[3]National Centre for Atmospheric Science, University of York, Heslington, York, YO10 5DD, UK.
[4]School of Geography, Earth and Environmental Sciences, University of Birmingham, Birmingham, B152TT, UK.

[a]now at: School of Environmental Science and Engineering, Nanjing University of Information Science and Technology, Nanjing, China
[b]now at: British Antarctic Survey, Natural Environment Research Council, Cambridge, CB3 0ET, UK

*Correspondence to*: Beth S. Nelson (beth.nelson@york.ac.uk)

**Abstract** In recent years, clean air policies have led to reductions in air pollution across China. Alongside this, emerging carbon neutrality (CN) policies that aim to address the impacts of climate change may also deliver air quality (AQ) co-benefits or climate penalties. Different CN policies will lead to different changes in volatile organic compound (VOC), $NO_x$, and particulate matter (PM) emissions, which will in-turn impact the photochemical production of secondary pollutants such as ozone ($O_3$). It is currently unclear how different combinations of AQ and CN policies may impact in situ $O_3$ production across China in the future. A detailed chemical box model incorporating the Master Chemical Mechanism was developed to investigate the impact of combined AQ and CN policies on $O_3$ formation in Beijing. The Multi-resolution Emission Inventory model for Climate and air pollution research (MEIC) and the Dynamic Projection model for Emissions in China (DPEC) were used to estimate future pollutant mixing ratios, relative to ambient observations of 35 VOCs, $NO_x$, CO and aerosol surface area (ASA) during the APHH-Beijing 2017 summer campaign. The most ambitious policy scenario, "*Ambitious Pollution 1.5D goals*", led to the largest reduction in $O_3$ production by 2060, but was not the most impactful scenario for reducing $O_3$ production between 2030-2045. Larger reductions were observed under the *"Ambitious Pollution Neutral goals"* policy which focuses on achieving net zero by 2060. $O_3$ production was found to be most sensitive to changes in the OLE2 group of VOCs (alkenes where $k_{OH} > 7 \times 10^4$ ppm$^{-1}$ min$^{-1}$; a 5% increase in OLE2 increased simulated $O_3$ production by 1.12%). However, reducing less reactive but higher concentration species in Beijing (including methanol and short-chain alkanes) led to larger reductions in $O_3$ production under all scenarios. $O_3$ production was not sensitive to changes in ASA, with a 69% decrease in ASA leading to a change of < 1% in $O_3$. However, doubling biogenic VOCs in the model further increased $O_3$ production in 2060 under all future scenarios by up to 18%, indicating that the influence of future climate-induced changes in biogenic emissions may have a significant impact on in situ $O_3$ formation in Beijing. This study highlights that the emission trajectories of certain specific VOCs are highly influential in determining possible future $O_3$ air quality effects that may arise from increasing ambient temperatures and decarbonisation in Beijing.

## 1 Introduction

Ground-level ozone ($O_3$) is a secondary air pollutant, formed from the photochemical reactions of volatile organic compounds (VOCs) and $NO_x$ (NO + $NO_2$) emitted into the atmosphere. $O_3$ can be detrimental to human health, mainly impacting the respiratory system. The physical effects of $O_3$ on the respiratory system include damage and inflammation to the airways, as well as decreased lung function. Epidemiological studies have also linked high $O_3$ to cardiovascular mortality, the impairment of cognitive development, and reproductive health.(Chen & Schwartz, 2009; Sharkhuu et al., 2011; World Health Organization, 2013). As well as being directly harmful to human health, ground-level $O_3$ can lead to crop degradation, which



can in turn lead to large economic losses. This can further impact human health when low crop yields increase food prices, a
particularly important issue in developing countries, and lead to malnourished populations. (Ainsworth, 2017; IPCC, 2015;
Mills et al., 2018) To reduce the global burden of ground-level $O_3$, effective pollution mitigation policies are required to reduce
emissions of chemical species leading to $O_3$ production. It is currently unclear how carbon neutrality and other climate targets
will impact $O_3$ precursor emissions, and subsequently $O_3$ production, in parallel with local air quality improvement
interventions.


Between 2010 and 2017, a significant decline in anthropogenic emissions of pollutants has been observed in China, owing to
the implementation of widespread controls. (Zheng et al., 2018) Despite an overall increase in population and GDP, reductions
in emissions of $NO_x$ (17%), CO (27%), $PM_{2.5}$ (35%), $PM_{10}$ (38%), BC (27%) and $SO_2$ (62%) have been estimated, with the
largest reductions achieved since 2013 following the implementation of the Clean Air Act.(J. Huang et al., 2018; Zheng et al.,
2018) Despite success in reducing transportation and domestic stove emissions, persistent growth in emissions from the
industry sector and solvent use means that non-methane VOC emissions have only slowed since 2013. (M. Li et al., 2019)
Between 2013 – 2017, ground-level concentrations of $O_3$ continued to increase due to secondary chemical processes, where
the reduction of a set of precursor species without a co-reduction in others may lead to increased $O_3$ formation. (J. Huang et
al., 2018) Ambient $O_3$ levels in China regularly exceeds health limits set by the World Health Organisation (WHO) in urban
centres and may continue to do so without the application of effective policy interventions.

Alongside direct air quality controls, future climate and carbon net-zero policies will also have consequences on secondary
pollutant formation, with $O_3$ one of the most difficult pollutants to predict for the future. Although climate policies are generally
anticipated to deliver significant co-benefits to air quality, there will likely also be some ground level $O_3$ climate penalties to
be account for.(Fu & Tian, 2019) Some interventions may to lead to disbenefits for air quality particularly for pollutants formed
through secondary and/or non-linear chemical processes. (Shindell & Smith, 2019).

The complexity of ground-level photochemical $O_3$ formation makes policy evaluation of climate intervention a challenge.
Ground-level $O_3$ is formed in urban areas from OH radical-initiated reactions with volatile organic compounds (VOCs) and
$NO_x$. Peroxy radical species ($\Sigma RO_2 + HO_2$), formed from reactions of VOCs with OH, oxidise ambient NO to $NO_2$. The rapid
daytime photolysis of $NO_2$ leads to the regeneration of NO, along with an oxygen atom ($O(^3P)$) which further reacts with $O_2$
to form $O_3$. $O_3$ production is sensitive to VOC structure and reactivity where sometimes low concentrations of highly reactive
VOCs may lead to significant $O_3$ production.(Nelson et al., 2021) However, when concentrations of $NO_x$ are very high relative
to VOCs, OH radicals preferentially react with $NO_2$, forming $HNO_3$, terminating the radical propagation cycle. This leads to
reduced peroxy radical formation, and thus a reduction in $O_3$ production and is often described as a VOC-limited regime. The
non-linear nature of photochemical $O_3$ production means that reductions in $NO_x$ under VOC-limited conditions may lead to
increased $O_3$ concentrations in $NO_x$-saturated urban areas. In areas of high $NO_x$, NO can act to remove $O_3$ ($O_3$ titration).
Consequently, reducing $NO_x$ could lead to higher ambient $O_3$ concentrations. This has recently been observed during the
SARS-CoV-2 pandemic, where lockdowns in urban centres led to reduced road traffic, and reduced $NO_x$ emissions that often
coincided with increased $O_3$ levels.(Lee et al., 2020; Sicard et al., 2020) In contrast, reductions in $NO_x$ in $NO_x$-limited
environments where $O_3$ production is controlled by changes in $NO_x$ lead to a reduction in $O_3$ production. Recent studies have
also shown that reducing particulate matter may exacerbate $O_3$ production further. This is due to the removal of a radical sink
process, whereby aerosol particles engage in $HO_2$ uptake. A recent modelling study over Asia described an "aerosol-inhibited"
chemical regime, where $O_3$ production can be significantly supressed by the presence of this heterogeneous radical sink.(Ivatt
et al., 2022)



Where comprehensive VOC datasets are available, observationally constrained models incorperating detailed chemical mechanisms, such as the Master Chemical Mechanism (MCM, mcm.york.ac.uk, (Jenkin et al., 2015)) can be used to investigate how changes in VOCs, $NO_x$ and aerosol surface area (ASA) may impact in situ $O_3$ formation. The sensitivity of in situ $O_3$

production rates to changes in individual species (e.g. isoprene) and species subset (e.g. alkanes, alkenes, aromatics) concentrations can then be explored. Alongside its photochemical production, ambient $O_3$ concentrations are also influenced by pollutant transportation both into and out of the measurement site. Although chemical box models do not account for pollutant transport or regional emissions, they are a useful tool to assess local $O_3$ production (Nelson et al., 2021; Whalley et al., 2018, 2021)


A measurement suite of $NO_x$, VOCs, CO, $SO_2$, HONO, aerosol concentration and size, photolysis rates and meteorological parameters was obtained during a measurement campaign in Beijing, China in the summer of 2017 as part of the Air Pollution and Human Heath (APHH) programme. (Shi et al., 2019) This comprehensive observational dataset is used in this study to tailor a chemical box model to simulate the instantaneous rates of $O_3$ production (base model). The APHH 2017 $NO_x$, VOC

and aerosol observations were then multiplied by scaling factors (section 2.4) to investigate the change in $O_3$ production rate under different scenarios. The scaling factors were obtained by comparing changes in emissions between the 2017 Multi-resolution Emission Inventory (MEIC), and future anthropogenic emission projections from the Dynamic Projection model for Emissions in China (DPEC) inventory up to 2060. The emissions taken from the DPEC inventory describe six scenarios, based on different air quality and carbon neutrality policies, and provide insight into how $O_3$ production may vary in the future.


## 2. Methodology

### 2.1 Site and instrument description

Ground-level measurements of volatile organic compounds (VOCs), $NO_x$, CO, $SO_2$, HONO, and aerosol concentration and size were made at the Institute of Atmospheric Physics (IAP), Chinese Academy of Sciences, during the summer of 2017. The

measurements were part of a larger Air Pollution and Human Health (APHH) programme, with the aim to better understand the sources, atmospheric transformations, and health impacts of air pollutants in Beijing. (Shi et al., 2019) The measurement site was located between the third and fourth northern ring roads in Beijing, in an urban residential area (39°58'33" N 116°22'41"E). A more detailed description of the measurement site and the instrumentation used can be found in Shi et al., 2019.


Speciated VOC measurements were obtained using a combination of PTR-ToF-MS (proton transfer reaction time-of-flight mass spectrometry) and a dual column-GC-FID instrument, coupled to Markes International CIA Advantage and Unity 2 systems for sampling and subsequent pre-concentration of ambient VOC species. Water was removed from the sample via a cold glass finger (-30°C), before it was adsorbed onto a Markes International Ozone Precursor dual-bed sorbent cold trap.

After pre-concentration, the sample was thermally desorbed onto a gas chromatograph, and split 50:50 onto two columns. This allowed for the detection of both oxygenated (50 m × 0.53 mm LOWOX column) and non-oxygenated (10 m × 0.53 mm $Al_2O_3$ PLOT column) species. A full list of observed VOCs used in this study is presented in Whalley et al., 2021, and summarised in section 3.3, Table 2.

### 2.2 Model description

A zero-dimensional chemical box model, incorporating a subset of the Master Chemical Mechanism (MCM v3.3.1; Jenkin et al., 2015; Saunders et al., 2003) into the open source AtChem2 modelling tool (Sommariva et al., 2020) was used to calculate



in situ $O_3$ production rates resulting from constraining to ambient concentrations of precursor species. The MCM is a near explicit description of the atmospheric chemical degradation of 143 VOCs, through 17,500 reactions of 6,900 species

(mcm.york.ac.uk; last access: March 2022). The model was constrained to the campaign averaged diel concentrations of 35 VOC species (Section 3.3, Table 2), $NO_x$, CO, $SO_2$, 9 photolysis rates, temperature, pressure, and relative humidity. Measured species were averaged or linearly interpolated to 15 min data before incorporation into the model. Each model was run for 5 d, with each day being constrained to the diel of the campaign averaged observations, or an adjustment of these observations to investigate sensitivities to $O_3$ production. Only the fifth day was used in this analysis to allow for the spin-up of model

generated intermediate compounds.

The model was also constrained to observed HONO, adjusted to a surface concentration to account for the vertical profile. This was calculated using campaign and hourly averaged measurements of the Deardorff velocity (w*), obtained during co-occurring flux measurements from a 325 m tower located next to the ground-level measurement site.(Shi et al., 2019) The

Deardorff velocity was used to approximate the rate of vertical HONO transport, allowing for the calculation of adjusted HONO ([HONO]$_{adj}$) from observed HONO ([HONO]$_{meas}$) using Eq. 1 (Nelson et al., 2021):

$$[HONO]_{adj} = [HONO]_{meas} \times e^{-j(HONO)t},\qquad(1)$$

where $t$ is the time taken for $[HONO]_{meas}$ to diffuse to the midpoint of the boundary layer at the measured Deardroff velocity (w*).


Total aerosol surface area (ASA) was also constrained in the model, to account for the effect of $HO_2$ aerosol uptake. The first order loss of $HO_2$ ($k$) to ASA was calculated using Eq 2:

$$k = \frac{\omega A \gamma}{4},\qquad(2)$$

Where $\omega$ is the mean molecular speed of $HO_2$ (43 725 cm s$^{-1}$ at 298 K), $\gamma$ is the aerosol uptake coefficient ($\gamma$ = 0.2, as

recommended by (Jacob, 2000), and A is the measured ASA in cm$^2$ cm$^{-3}$.

The model was constrained to the measured photolysis frequencies of $j(O(^1D))$, $j(NO_2)$ and $j(HONO)$, calculated from the measured wavelength-resolved actinic flux, as well as published absorption cross-sections and photodissociation quantum yield data (IUPAC Task Group, https://iupac.aeris-data.fr/, last access: September 2023). The photolysis rates of $NO_3$, HCHO, $CH_3CHO$ and $CH_3COCH_3$ were calculated by scaling the ratio of clear-sky $j(O1D)$ or $j(NO_2)$ to observed $j(O^1D)$ or $j(NO_2)$

rates, depending on whether the species photolyses above or below 360 nm. A full description of this methodology can be found in Whalley et al., 2021.

The physical loss of model generated species through deposition or ventilation was estimated by running a model to produce glyoxal and varying these loss rates. Observed glyoxal concentrations were reproducible by the model when the model was constrained to measured boundary layer depth, and a deposition velocity of 0.5 cm s$^{-1}$ was applied, along with a ventilation

term as described by Whalley et al., 2021. A comparison of measured and modelled glyoxal showing good agreement is presented in the supplementary (Figure S1).

### 2.3 $O_3$ production and formation potential calculations

The rate of in situ $O_3$ production was calculated by subtracting the rate of instantaneous $O_3$ loss $L(O_3)_{inst}$ from the rate of

instantaneous $O_3$ production $P(O_3)_{inst}$ (see Eqs. 3 - 5). $P(O_3)_{inst}$ is the sum of reactions of NO with both $HO_2$ and $RO_2$, leading



to $NO_2$ formation. $L(O_3)_{inst}$ is the sum of all $O_3$ loss rates, along with any routes of $NO_2$ consumption that do not lead to $O_3$ formation. This includes $O_3$ loss to $HO_2$, and to water via $O^1D$, as well as $NO_2$ loss to OH and $RO_2$.

$$P(O_3)_{inst} = \left( k_{HO_2+NO}[HO_2][NO] + \sum_i k_{RO_{2_i}+NO}[RO_2][NO] \right), \tag{3}$$


$$L(O_3)_{inst} = j(O^1D)[O_3] * f + k_{OH+O_3}[OH][O_3] + k_{HO_2+O_3}[O_3][HO_2] + k_{OH+NO_2+M}[NO_2][OH][M] +$$
$$\sum_i k_{RO_{2_i}+NO_2+M}[RO_2][NO_2][M], \tag{4}$$

where $f$ is the fraction of $O(^1D)$ atoms that react with water vapour to form OH, rather than undergoing collision stabilisation.

The overall net production of $O_3$. $P(O_3)$ was then calculated using Eq. 5:

$$P(O_3) = P(O_3)_{inst} - L(O_3)_{inst}, \tag{5}$$

$O_3$ formation potentials (OFPs) were calculated for each group of VOCs (Table 2) using Eq. 6 (Z. Liu et al., 2023)

$$OFP_i = [VOC]_i \times MIR \tag{6}$$

Where MIR is the Maximum Incremental Reactivity of each VOC, taken from Carter, 2010, and [VOC] is the concentration of an individual VOC in µg m$^{-3}$. The MIR is a metric commonly used to measure the photochemical reactivity of an individual VOC by estimating the mass of $O_3$ produced per fixed mass of the compound of interest (Carter, 2010). The values taken from Carter, 2010 are applicable to VOC-limited regions with high NO concentrations. For groups of VOCs (e.g. ARO1, see Table 1), the sum of the OFPs (ΣOFP) is calculated by determining the average daytime (05:00 – 15:00 local time) OFP for each

individual VOC, and then summing these OFPs.

### 2.4 Model scenarios

The Multi-resolution Emission Inventory model for Climate and air pollution research (MEIC) and the Dynamic Projection model for Emissions in China (DPEC) were used to derive estimates for future ambient mixing ratios of chemical species, relative to observations made during the APHH Beijing summer campaign (http://meicmodel.org.cn/, last access: June 2022).

The MEIC inventory provides a high-resolution (0.25°) data base of anthropogenic emissions of air pollutants.(M. Li et al., 2014, 2017, 2019; Zheng et al., 2018) The DPEC inventory aims to dynamically project the future emissions of air pollutants in China in the context of socio-economic development, global climate adaptations, carbon neutrality targets, combining pollution mitigation policies and carbon reduction pathways (Cheng et al., 2021). The DPEC inventory contains projections

every five years until 2060 for six different air pollution mitigation and carbon neutrality scenarios, described in Table 1. Averaged inventory data for May and June was used in this study, aligning with the APHH-2017 summer campaign.

Future estimations of VOCs, $NO_x$ and aerosol surface area (ASA) were determined by multiplying the 2017 observed values by a scaling factor, calculated by comparing anthropogenic emissions in the 2017 MEIC inventory to emissions in the DPEC

inventory for the Beijing region. In this study, all compounds were assumed to have lifetimes within the footprint of the inventory grid. However, we acknowledge that the observed concentrations of some compounds with longer lifetimes (e.g. ethane, propane) will be impacted by emission changes further afield.

The APHH 2017 observational dataset was adjusted in five ways to investigate changes to in situ $O_3$ production.



**Scenario 1:** Anthropogenic VOCs (see Table 2), which includes all observed VOCs excluding isoprene, α-pinene, and limonene (defined hereafter as AVOCs) and $NO_x$ were each multiplied by a scaling factor of 0.01, 0.025, 0.05, 0.075, 0.1, 0.2, 0.25, 0.3, 0.4, 0.5, 0.6, 0.7, 0.75, 0.8, 0.9, 1, 1.1, 1.25, 1.5, 1.75, 2. The model was run for all 441 combinations of these factors for VOCs and $NO_x$. This allowed for isopleths to be constructed and, hence the identification of $NO_x$-limited / VOC-limited transition points.

**Scenario 2**: AVOCs, $NO_x$, and ASA were varied using scaling factors calculated from the difference between emissions defined in the 2017 MEIC inventory, and projected emissions under six future DPEC scenarios every 5 years between 2025 and 2060 (Table 1).

**Scenario 3**: AVOCs and $NO_x$ were varied using scaling factors calculated from the difference between emissions defined in a 2017 inventory, and projected emissions under the six future DPEC scenarios, but ASA remains fixed at 2017 APHH observed levels.

**Scenario 4**: AVOCs were grouped as defined in the SAPRC07 mechanism (section 3.3, Table 2) and incrementally increased by 5% to assess in situ $O_3$ sensitivity to individual VOC groupings. A description of the SAPRC07 speciation can be found in (Carter, 2010).

**Scenario 5**: The observed mixing ratios of VOCs likely to be predominantly from biogenic sources (isoprene, α-pinene, and limonene) are multiplied by a factor of 1, 1.1, 1.2, 1.3, 1.4, 1.5, 1.6, 1.7, 1.8, 1.9 and 2 for six future DPEC scenarios for 2060. AVOCs are varied as per DPEC projections.

**Table 1**: Future scenarios based on different climate and air quality outcomes, as defined by the Dynamic Projection Model for Emissions in China (DPEC). A detailed description of these scenarios can be found in Cheng et al., 2021.

| Scenario | Climate constraints | Socioeconomic drivers | End-of-pipe pollution control |
|---|---|---|---|
| Baseline | RCP6.0 | SSP4 | Same as 2015 levels. |
| Current goals | RCP4.5 | SSP2 | Current released and upcoming policies. |
| Ambitious pollution NDC goals | RCP4.5 | SSP2 | Best available end-of-pipe pollution control technologies. |
| Ambitious pollution Neutral goals | China's net zero $CO_2$ emissions in 2060* | SSP1 | |
| Ambitious pollution 2D goals | RCP2.6 | SSP1 | |
| Ambitious pollution 1.5D goals | RCP1.9 | SSP1 | |

*This falls between RCP1.9 and RCP2.6.






## 3 Results and Discussion

### 3.1 Projected changes in bulk VOCs and NO$_x$ since 2017 observations

The DPEC emissions inventory was used to project changes in anthropogenic VOCs and NO$_x$ across Beijing since the 2017 APHH measurement campaign as described in section 2.4. Projected changes in anthropogenic VOCs and NO$_x$ were estimated

by comparing emissions estimate differences from the MEIC 2017 inventory with respective DPEC inventories. Figure 1 shows the projected percentage change in anthropogenic VOCs and NO$_x$ up to 2060 for the six future scenarios defined in Table 1.

Under current pollution and climate goals defined by DPEC, where intermediate air quality and climate policies are applied

(*Current goals*), NO$_x$ and VOCs are projected to reduce by *ca.* 48% and 31% in the Beijing region respectively, compared to 2017 observed values, by 2030. Reductions in NO$_x$ and VOCs are projected for all air quality and climate policy scenarios, except for the *Baseline* scenario. The largest reductions in NO$_x$ are achieved with *Ambitious Pollution Neutral* goals (59%), followed by *Ambitious Pollution 1.5D* and *2D* goals (both 56%). The largest reductions in VOCs are predicted with *Ambitious Pollution 1.5D* goals (42%), closely followed by *Ambitious Pollution Neutral* goals (41%) and *Ambitious Pollution 2D* goals

(40%). However, projections for the *Baseline* scenario show increased NO$_x$ and VOCs by 2030 (43% and 13% respectively).

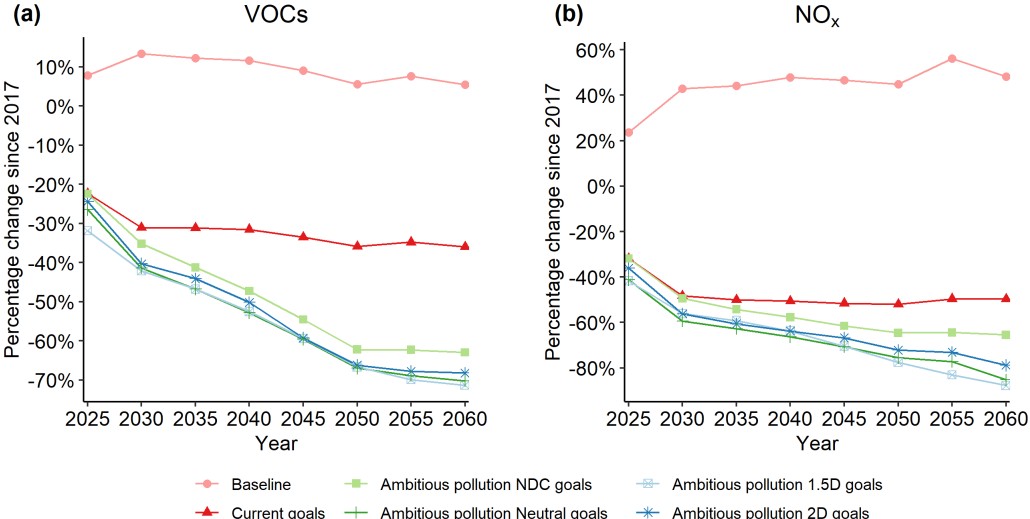

**Figure 1:** Projected change in total anthropogenic VOCs (a) and NO$_x$ (b) emissions for the Beijing region since the 2017 APHH Beijing summer campaign. Six future scenarios described in the DPEC inventory are presented up to 2060.


### 3.2 The effect of changes in observed AVOCs and NO$_x$ on modelled in situ O$_3$ production rates

First, a model constrained to observations of AVOCs and NO$_x$ during APHH-Beijing was varied by different scaling factors to investigate the sensitivity of in situ O$_3$ production rates to its precursor species (scenario 1, section 2.4). This variation is not related to the DPEC scenarios. The resulting isopleth indicates that O$_3$ production at the measurement site in 2017 was

VOC-limited (see black diamond, Figure 2), consistent with previous studies e.g. (Q. Li et al., 2020; Ren et al., 2022) Reductions in NO$_x$ up to *ca.* 75% without a co-reduction in total VOCs lead to an overall increase in daytime in situ O$_3$ production rate of up to 30%. When NO$_x$ is reduced by more than *ca.* 75%, there is a switch in chemical regime to become





NO$_x$-limited, and further reductions in NO$_x$ lead to reductions in O$_3$ production. The scaled changes in NO$_x$ and VOCs under the six DPEC scenarios (Table 1) were then mapped onto to the modelled O$_3$ production rates calculated independently to the

DPEC projections (scenario 1, section 2.4). The position of each DPEC scenario on the modelled P(O$_3$) isopleth in 2030 and 2060 using projections for NO$_x$ and total bulk AVOCs are shown by the filled diamonds in Figure 2.

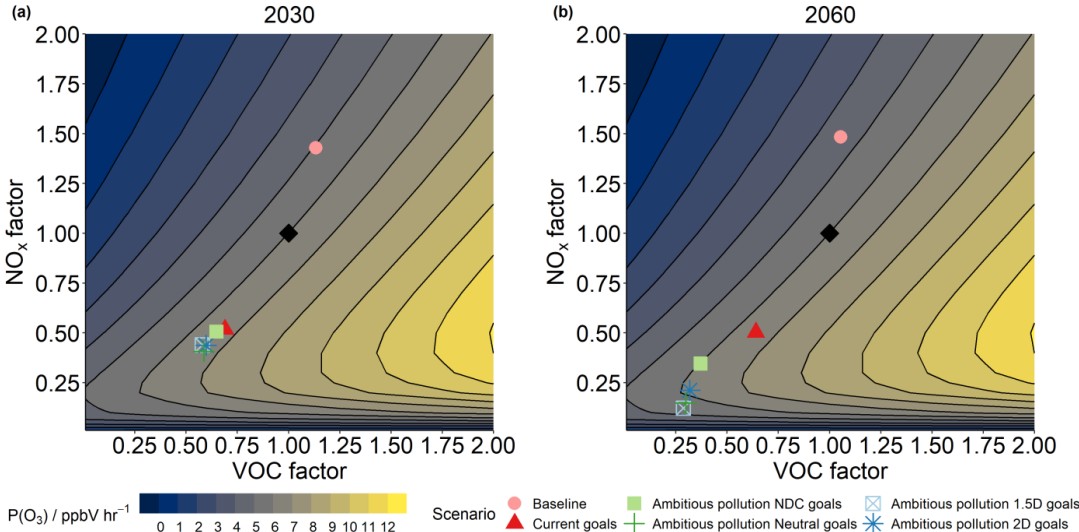

**Figure 2:** Change in daytime (05:00 – 15:00 local time) O$_3$ production rates arising from varying observational NO$_x$ and VOCs
by a scaling factor independent of the DPEC inventory. The black diamond represents modelled O$_3$ production rates using observed values during the APHH 2017 campaign. The coloured shapes represent the modelled O$_3$ production rates for each DPEC scenario (Table 1), determined from the projected change in NO$_x$ and VOC concentrations by 2030 (a, left) and 2060 (b, right) using the DPEC inventory.

Excluding the *Baseline* scenario, all scenarios show slight increases in local O$_3$ production rate by 2030 compared to 2017 observations (*ca.* 0.5 ppb h$^{-1}$). In contrast, the O$_3$ production rate decreases under the *Baseline* scenario by *ca.* 1 ppb h$^{-1}$, despite increasing NO$_x$ and VOCs. By 2060, projected O$_3$ production rates are comparable to the 2017 rate for all ambitious pollution policy scenarios, with a 0.5 ppb h$^{-1}$ increase estimated for the *Current goals* scenario. O$_3$ production rates have reduced further for the *Baseline* scenario, by *ca.* 1.5 ppb h$^{-1}$ compared to 2017 rates, by a further 0.5 ppb h$^{-1}$ compared to 2030. 2060 scenario
NO$_x$ reductions are sufficient that all ambitious pollution policy scenarios are approaching a transition from a VOC-limited to a NO$_x$-limited chemical regime, where further reductions in NO$_x$ without additional VOC reduction interventions would result in further reductions in O$_3$ production.

The above analysis provides an overall trend that in situ O$_3$ production might be expected to change when bulk AVOCs are
varied alongside NO$_x$. However, we recognize that varying NO$_x$ and bulk AVOCs by a fixed scaling factor is not necessarily representative of dynamic changes across multiple sources/sectors. As different policies apply different air pollution and climate change abatement strategies, the types and ratios of different VOC emission changes under different policies may vary significantly. In addition, calculated O$_3$ production rates may not directly translate into increased or decreased ambient O$_3$ levels in the future since overall O$_3$ levels are also dependent on regional effects and chemical transportation. Despite this, the
chemical modelling of in situ O$_3$ can still give us important insights into which chemical species have the greatest impact on in situ O$_3$ production, which contributes to overall local O$_3$ levels.





### 3.3 The effect of changes in VOC groups on in situ O₃ production rate

To investigate how in situ $O_3$ production rate varies under different Carbon Neutrality (CN) policies (section 2.4, Table 1), $O_3$
production rates under each scenario were modelled, up to 2060. Rather than varying bulk VOCs, grouped projected AVOCs
were calculated using the DPEC defined change in emissions of the VOC. The VOC groupings are as defined by those in the
SAPRC07 chemical mechanism (Carter, 2010; M. Li et al., 2014), as outlined in Table 2. Emission projections for the different
future scenarios are highly uncertain due to their socioeconomic and political nature, but can provide us some insight into how
in situ $O_3$ production might be impacted by different variations in its precursor species.


**Table 2**: List of volatile organic compounds included in box modelling analysis, including measurement instrument and
SAPRC07 group speciation.

| SAPRC07 Group | Volatile organic compound | Instrument |
|---|---|---|
| ALK1 | ethane | DC-GC-FID |
| ALK2 | propane | (Hopkins et al., 2011) |
| ALK3 | *n*-butane, *i*-butane, ethanol | |
| ALK4 | *n*-pentane, *i*-pentane, (2&3)-methylpentane, *n*-hexane, *n*-heptane, | |
| ALK5 | *n*-octane | |
| ETHE | Ethene | |
| OLE1 | propene, but-1-ene, pent-1-ene | |
| OLE2 | cis-but-2-ene, trans-but-2-ene, methylpropene, trans-pent-2-ene, 1,3-butadiene | |
| ACYE | acetylene | |
| CCHO | acetaldehyde | |
| MEOH | methanol | |
| BENZ | benzene | |
| ISOP* | isoprene | |
| ARO1 | toluene, *i*-propylbenzene, *n*-propylbenzene | DC-GC-FID (toluene, xylenes) |
| ARO2 | 1,2,3-trimethylbenzene, 1,2,4-trimethylbenzene, 1,3,5-trimethylbenzene, (m,p)-xylene, o-xylene | PTR-ToF-MS (propylbenzenes, trimethylbenzenes) |
| MACR | methacrolein | PTR-ToF-MS (Z. Huang et al., 2016) |
| MVK | methylvinylketone | |
| TERP* | α-pinene, limonene | |
| HCHO | formaldehyde | LIF (Cryer, 2016) |

* Mixing ratios of ISOP and TERP are not included in "AVOCs" and are kept constant when VOCs are varied by the DPEC
inventory, since they are considered to derive mainly from biogenic sources.


The projected changes in mixing ratio for a selection of AVOCs under the six different scenarios are presented in Figure 3. By
2025, there is an overall reduction in AVOCs (as defined by Table 2, excluding isoprene, α-pinene and limonene which remain
at APHH-2017 levels) of between 25-33%, except for the *Baseline* scenario where AVOCs remain unchanged (<1%). For
many of the reactive AVOCs, large reductions are expected between 2017 and the first projected future scenario in 2025 under





all air quality and climate change scenarios. These large reductions (Fig. 3) can be seen in the ARO1 and ARO2 groups (*ca.* 96% reduction), as well as the OLE1, OLE2, ALK3 and ALK4 groups (*ca.* 75-95% reduction). Other VOCs groups, such as ACYE, ALK1 and ETHE initially increase under all scenarios up to 2025 (by *ca.* 35-65%), before a gradual decline up to 2060 under ambitious pollution scenarios (by *ca.* 50-70%). Overall, the largest VOC reductions are observed under the most ambitious air quality and climate change scenario, *Ambitious Pollution 1.5D* goals (by 59%). However, there are some notable

exceptions. High concentrations of methanol (MEOH) were observed in the APHH-2017 Beijing campaign and the DPEC inventory does not forecast MEOH to reduce as much in the *Ambitious Pollution 1.5D* goals scenario, as it does in the other ambitious pollution scenarios between 2025 and 2060. In addition, the ALK1 group is estimated to increase in the region by *ca.* 19% by 2040 under the *Ambitious Pollution 15D* scenario but reduces or remains unchanged under the other ambitious pollution scenarios.

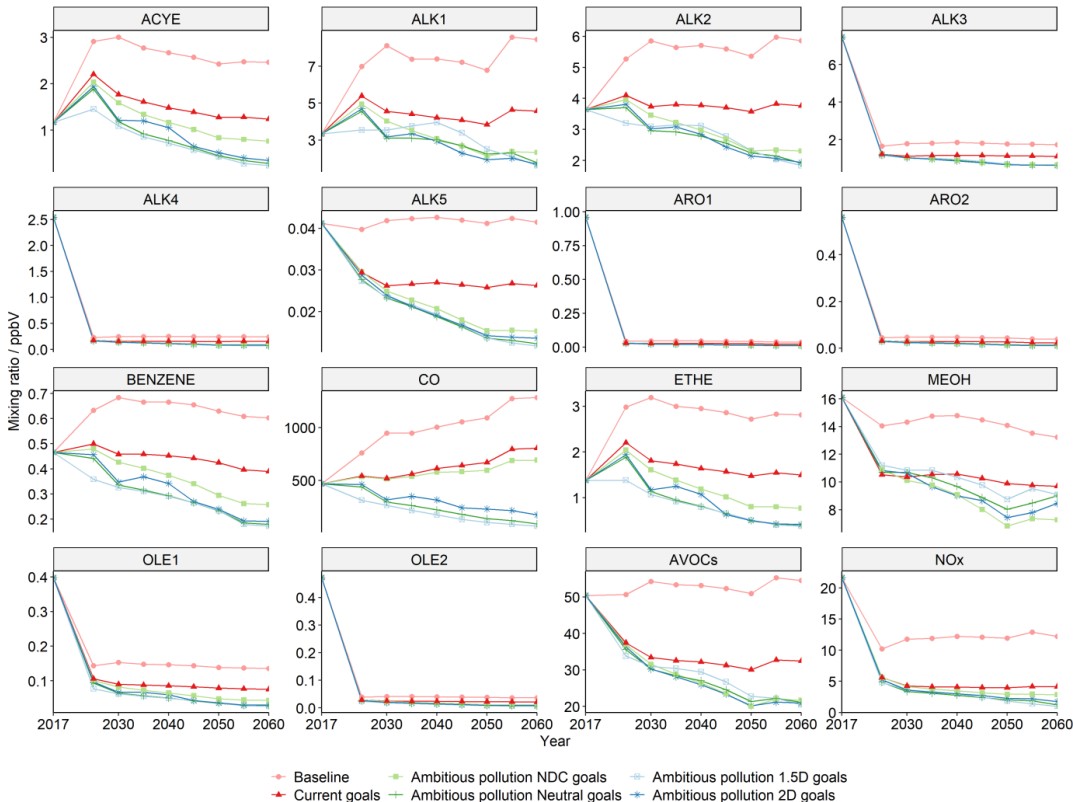


**Figure 3:** Projected absolute change in mixing ratio of key AVOC sub-groups (those observed during APHH-2017) and NO$_x$ for the six DPEC air pollution and climate policy scenarios (Table 1) every 5 years between 2025 and 2060, in comparison to APHH-2017 Beijing campaign. AVOC includes all VOCs observed during APHH 2017, excluding isoprene, α-pinene, and limonene (Table 2). Note that the y-axis scale is different on each sub-plot.


The resulting modelled O$_3$ production rates when DPEC scaled changes to concentrations of NO$_x$, SAPRC07 speciated AVOCs, CO and ASA are applied is presented in Figure 4 (scenario 2, section 2.4).





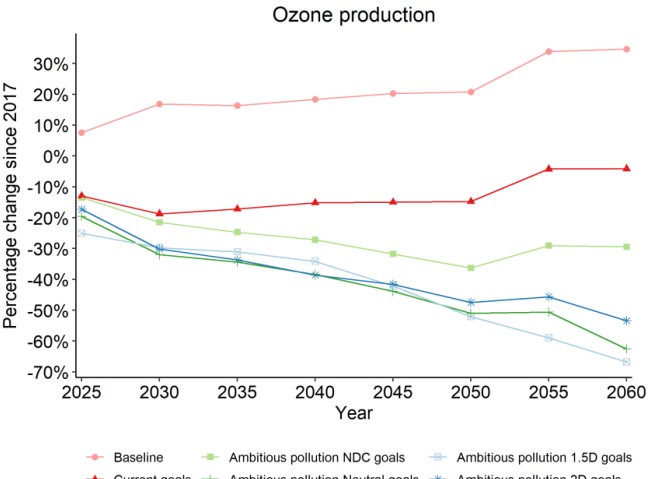

**Figure 4:** Projected percentage change in $O_3$ production rate since 2017 observations, when VOC and $NO_x$ observations are scaled using the DPEC emissions inventory.

In contrast to the isopleth projections presented in Figure 3, when the subset of anthropogenic SAPRC07 speciated VOCs and $NO_x$ are varied, overall all modelled $O_3$ production rates decrease up to 2060 under all scenarios except for the *Baseline* scenario (by *ca.* 30-60%). When minimal air quality and climate policies are applied (*Baseline* scenario), $O_3$ production rate steadily increases up to 2050 by *ca.* 20%. Between 2050 and 2055, a steep increase of a further *ca.* 15% is projected. Under all other scenarios, $O_3$ production rate reduces with time. However, under the *Current goals* scenario, the reduction in $O_3$ formation is significant at first (*ca.* 20% by 2030) but increases after 2050 resulting in a minimal difference in production rates between 2017 and 2060. More significant reductions in $O_3$ production are observed under the ambitious pollution and climate policy scenarios. The largest reduction in $O_3$ production rate by 2060 is observed under the Ambitious pollution policy combined with a climate policy limiting global warming to +1.5 °C, *Ambitious pollution 1.5D goals* (ca. 60%). However, this policy is not the most effective at reducing local $O_3$ production rates during intermediate years. Interestingly, between 2030 and 2045, two ambitious pollution scenarios with less rigorous climate policies lead to marginally larger reductions in $O_3$ production rate. Ambitious pollution policies with a global warming limit of +2 °C (*Ambitious Pollution 2D* goals), and with a net zero policy by 2060 (*Ambitious Pollution Neutral* goals), both reduce $O_3$ production further than the most ambitious climate policy scenario between 2030 and 2045.

The contrast in results for all scenarios using the isopleth analysis (Figure 2) and the scenario modelling study (Figure 3) alongside the variation in bulk VOC and $NO_x$ in the different scenarios (Figure 1), highlights the importance of using a detailed chemical mechanism to investigate how changes in emissions might impact $O_3$ production rates in future scenarios. When $NO_x$ and VOCs are varied in bulk, the only scenario projected to reduce $O_3$ production is the *Baseline* scenario. However, the opposite is true when speciated AVOCs are varied by projections defined by the SAPRC07 AVOC groupings in the DPEC inventory. The different scenarios result in different emission reductions in different AVOCs, with different propensities to lead to $O_3$ production.






**3.5 Sensitivity of changes in concentration of specific AVOCs and ASA on in situ O₃ production**

To further investigate which chemical species are driving the largest changes in $O_3$ production rate, SAPRC07 AVOC groups were incrementally increased by 5%, with all other groups remaining at 2017 observed levels. The resulting change in $O_3$ production rate for each incremental change is presented in Table 3, and compared to the group determined Maximum

Incremental Reactivities (MIR) in high $NO_x$ conditions, and the subsequently calculated $O_3$ formation potentials (OFP) (see section 2.3).

**Table 3:** Change in O₃ production rate, P(O₃), on incrementally increasing the observed concentrations of each SAPRC07 AVOC grouping by +5%. Changes in P(O₃) are listed in descending order from largest increase in P(O₃). These are compared

to MIRs determined by Carter, 2010, and the calculated sum of OFPs for each group. (Z. Liu et al., 2023)

| Group | Volatile organic compound | ΔP(O₃) / % | MIR (mean) | Σ(OFP) / μg m⁻³ |
|---|---|---|---|---|
| OLE2 | cis-but-2-ene, trans-but-2-ene, methylpropene, trans-pent-2-ene, 1,3-butadiene | + 1.12 | 12.46 | 12.44 |
| ALK3 | n-butane, i-butane, ethanol | + 0.35 | 1.30 | 21.96 |
| ALK4 | n-pentane, i-pentane, (2&3)-methylpentane, n-hexane, n-heptane | + 0.32 | 1.40 | 11.08 |
| OLE1 | propene, but-1-ene, pent-1-ene | + 0.29 | 9.53 | 10.87 |
| ETHE | ethene | + 0.24 | 9.00 | 15.62 |
| ARO2 | 1,2,3-trimethylbenzene, 1,2,4-trimethylbenzene, 1,3,5-trimethylbenzene, (m,p)-xylene, o-xylene | + 0.23 | 9.31 | 20.26 |
| MEOH | methanol | + 0.17 | 0.67 | 14.39 |
| ARO1 | toluene, i-propylbenzene, n-propylbenzene | + 0.15 | 2.85 | 14.59 |
| ALK2 | propane | + 0.08 | 0.49 | 3.15 |
| ALK1 | ethane | + 0.02 | 0.28 | 1.20 |
| BENZENE | benzene | + 0.01 | 0.72 | 1.09 |
| ALK5 | n-octane | + 0.01 | 0.90 | 0.18 |
| ACYE | acetylene | < + 0.01 | 0.95 | 1.31 |
| ASA | aerosol surface area | - 0.12 | - | - |

Local in situ $O_3$ production rates in 2017 were most sensitive to changes in the OLE2 group ($\Delta P(O_3) = +1.12\%$), which includes highly reactive $C_4$-$C_5$ alkenes such as the but-2-enes and trans-pent-2-ene (Table 3). During APHH 2017, alkene concentrations were reported to be much higher than a comparable field campaign in London, with mean alkene concentrations more than

double that observed during the ClearFlo summer campaign in 2012 (Whalley et al., 2021). Higher concentrations observed during the 2017 campaign, combined with their fast reactivity ($k_{OH} > 7 \times 10^4$ ppm⁻¹ min⁻¹) of these alkenes results in a high sensitivity of $O_3$ production toward this group. After the OLE2 group, $O_3$ production was most sensitive to changes in the ALK3 and ALK4 groups, which include the $C_4$-$C_7$ alkanes and ethanol, followed by the OLE1 group which includes the less reactive alkenes ($k_{OH} < 7 \times 10^4$ ppm⁻¹ min⁻¹, excluding ethene) such as propene and but-1-ene.


The observed changes in $P(O_3)$ on increasing selected species by 5% are generally in agreement with the Maximum Incremental Reactivities (MIR) of each species, determined by Carter, 2010. However, the total $O_3$ formation potential ($\Sigma$OFP) of the ALK3 group is almost double that of the OLE2 group, despite $\Delta P(O_3)$ being three times more sensitive to a 5% increase in OLE2 than for ALK3. The large OFP attributed to ALK3 is explained by the very high concentrations of ethanol observed



during the APHH 2017 campaign. However, modelled P(O₃) is less sensitive to small changes in ethanol in the APHH-2017 model, where the full chemistry, bespoke to the observational data, is accounted for. Although the MIRs derived by Carter et al., 2010, and subsequently calculated ΣOFPs, are a good guide for determining the key contributors to O₃ formation, the detailed chemical model provides a more bespoke tool for assessing the key drivers of in siu O₃ formation at this particular location.


The sensitivities of modelled O₃ production to these VOCs can be combined with the trends in projected concentrations presented in Figure 4 to explain why O₃ production is not reduced the most under the *Ambitious Pollution 1.5D goals* between 2030 - 2045. Methanol emissions are projected to increase compared to 2017 levels under the *Ambitious Pollution 1.5D goals* until 2040 and are a large proportion of AVOCs by mixing ratio (Figure 3). Given that increasing methanol mixing ratios by

5% leads to an increase in O₃ production of 0.17%, the observed increase in methanol of *ca.* 25% up to 2040 could have a more pronounced impact on increasing O₃ production than smaller incremental changes in more reactive species such as OLE2, that are present in much smaller concentrations by 2025 (< 0.1 ppbV compared to *ca.* 10 ppbV of methanol). This is reflected in the high calculated OFP for methanol, compared to its MIR, which shows that the high concentrations of methanol observed results in this compound having a similar OFP to the sum of the more reactive ARO1 group. Similarly, although this study

shows that modelled O₃ production rate is not very sensitive to changes in the ALK2 group, substantial concentrations of this VOC group are observed. As a result, the overall trend in AVOCs (Table 2, excluding isoprene, α-pinene, and limonene) shows a weaker decline for the *Ambitious Pollution 1.5D* scenario which is reflected in the modelled O₃ production trend (Figure 5).

**3.6 The effect of changes in ASA on in situ O₃ production**

The observed aerosol surface area during APHH-Beijing 2017 campaign was also varied using DPEC projections for changes in PM$_{2.5}$ and PM$_{10}$. A more detailed description of how aerosol surface area (ASA) is incorporated into the chemical mechanism can be found in section 2.2. The future scenarios were modelled with and without DPEC derived variations in ASA (scenarios 2 and 3, section 2.4). DPEC projected ASA decreases with time under all scenarios, except for the Baseline scenario. Under Ambitious Pollution 1.5D, 2D and Neutral goals, ASA is projected to increase up to 2025, before decreasing by ca. 20-40%

of 2017 levels by 2030, and then further decreasing to 50-70% of 2017 values by 2060. ASA is projected to increase under Ambitious pollution NDC goals, Current goals and Baseline scenarios by 2030 (64%, 71%, and 250% respectively) and by 2060 (21%, 55%, and 221% respectively) relative to 2017 levels. Despite large percentage changes in ASA, very small changes in O₃ production rate are estimated when DPEC ASA estimates are applied in the model (Figure 6).




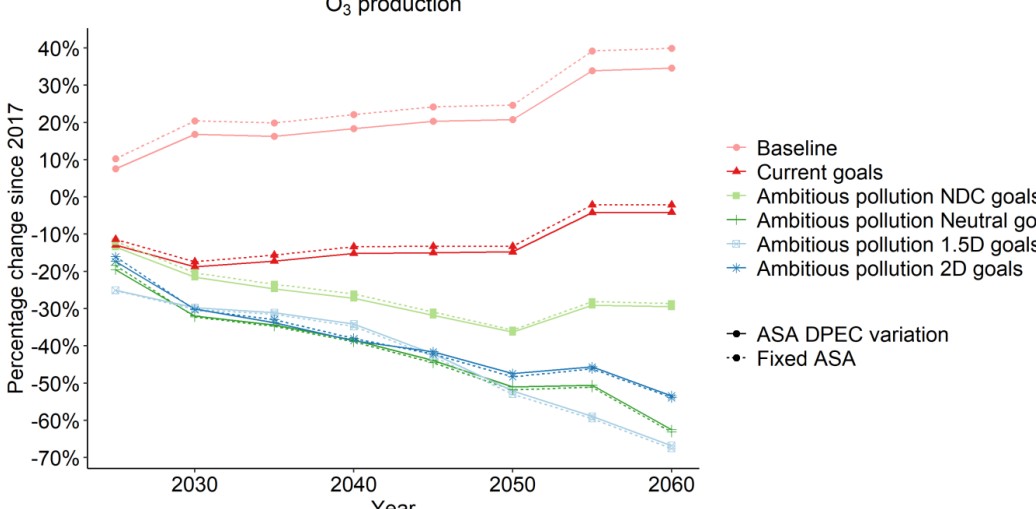

**Figure 6**: Modelled changes in $O_3$ production rate since 2017 levels under 6 different DPEC future scenarios between 2025 and 2060. Both the modelled $O_3$ production fixed at observed 2017 ASA levels (dashed line), and modelled $O_3$ production rate with ASA varying according to DPEC derived estimates (solid line) are shown.


On comparing $O_3$ production rate with or without the inclusion of the DPEC derived ASAs, the largest percentage difference in $O_3$ production was under the *Baseline* scenario, where $O_3$ production was 5.3% lower in 2055 and 2060 when the large increases in ASA since 2017 were applied (221%). This can also be attributed to the large concentrations of AVOCs projected

from the MEIC emissions estimates, leading to a larger supply of $HO_2$ radicals to be taken up by the ASA enhancement. Under the only other two scenarios where ASA was estimated to increase less steeply (*Current goals* and *NDC goals*, by 55% and 22% by 2060 respectively), including DPEC ASA values had very little impact (1-2% difference in $O_3$ production), and even smaller changes were found for scenarios where DPEC ASA was projected to decrease (*Ambitious pollution 1.5D*, *2D* and *Neutral goals*, by 69%, 59% and 64% by 2060 respectively), with percentage differences <1%. This suggests changes in ASA

will not appreciably impact $O_3$ production under the future DPEC scenarios. A study by Whalley et al., 2021 evaluating $O_3$ formation sensitivity to the APHH-2017 observations found that reductions in ASA enhanced $HO_2$ concentrations only under very low $NO_x$ (< 0.3 ppbV) at observed VOC levels This suggests that under the DPEC scenarios presented here, co-reductions in VOCs alongside $NO_x$ sufficiently reduce $HO_2$ to reduce the impact of ASA on resultant $O_3$ production rates.

**3.4 The effect of biogenic compounds on in situ $O_3$ production rate**

Whilst DPEC projections can be used to project changes in anthropogenic emissions of VOCs and $NO_x$, it is less clear how local biogenic emissions will change up to 2060. As compounds such as isoprene and the monoterpenes are primarily from biogenic sources, changes in their mixing ratios have not yet been accounted for in the anthropogenic VOC subset (AVOCs) used in this modelling study. However, with increasing global temperatures and urban greening, it is estimated that biogenic

VOC (BVOC) emissions in Beijing will increase over time, with recent studies estimating a 25% increase in biogenic emissions in China in the 2050s (S. Liu et al., 2019; Xie et al., 2017)




To investigate the sensitivity of projected $O_3$ production rate to increasing BVOC emissions under the six different scenarios, the models were re-run for 2060, with isoprene, α-pinene and limonene multiplied by a scaling factor of between 1 and 2 at 0.1 increments (scenario 5, section 2.4). Figure 7 shows how the change in $O_3$ production rate since 2017 varies for the six scenarios with increasing isoprene, α-pinene and limonene in 2060.

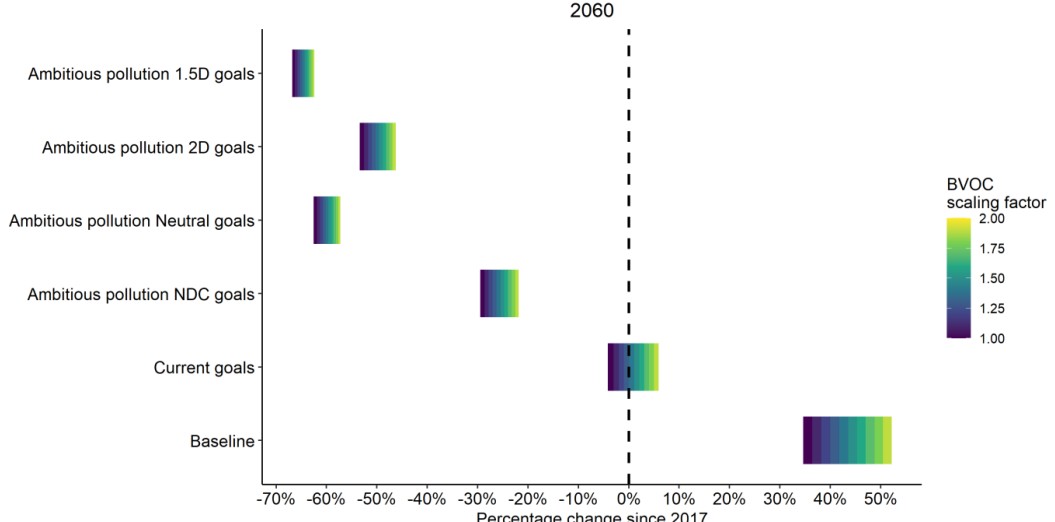

**Figure 7:** Percentage change in $O_3$ in 2060 since 2017 for the six different DPEC future scenarios (Table 1). Coloured bars show the range of percentage change when BVOC concentrations (isoprene, α-pinene, and limonene) are multiplied by a scaling factor of between 1-2. Values to the left and right of the dashed line indicate decreasing and increasing $O_3$ production rates respectively.

$O_3$ production rates calculated using the less ambitious *Baseline* and *Current goals* scenarios were found to be most sensitive to increasing biogenic concentrations in 2060. For the *Baseline* scenario, doubling biogenic concentrations led to a further 18% increase in $O_3$ production rates since 2017, compared to APHH-2017 concentrations (scaling factor = 1). In the *Current goals* scenario, a switch in the $O_3$ production rate from being a reduction to an increase since 2017 was found when biogenic concentrations were increased by *c.a.* 40%. In all ambitious scenarios, $O_3$ production rates are found to decrease since 2017, even when biogenic concentrations are doubled. However, the *Ambitious pollution 2D goals* and *Ambitious pollution NDC goals* are more sensitive to increasing biogenic concentrations (7% and 8% increase in percentage change in $O_3$ production rates on doubling respectively), compared to the *Ambitious pollution 1.5D goals* and *Ambitious pollution NDC goals* (4% and 5% respectively). In all cases presented here, < 2% of the sensitivity is attributed to limonene and α-pinene, and almost all the sensitivity observed here can be attributed to changes in isoprene alone. However, the impact of BVOC emissions on $O_3$ production rates is highly uncertain, and it is likely that there are many more fast reacting terpenes present in Beijing whose reactivity not accounted for in this study. In addition, from a pollution abatement perspective, increasing BVOCs will be much harder to controls than AVOCs. This sensitivity study highlights the importance of understanding the biogenic speciation and how biogenic compounds are expected to vary in the Beijing region, as these compounds are likely to have important implications on in situ $O_3$ production in this urban environment.



**4 Conclusions**

Future in situ $O_3$ production rates have been investigated for Beijing using detailed measurements of precursor species taken during the APHH 2017 summer campaign alongside future air quality and climate policy emission projections for the Beijing region. A chemical isopleth indicated a currently VOC-limited regime (in 2017), which would switch to a $NO_x$-limited regime

if $NO_x$ alone was reduced by ~75%. Based on this, and on estimated reductions in total VOCs and $NO_x$ under the DPEC scenarios, $O_3$ production rates were projected to increase under all four *Ambitious Pollution* scenarios. However, when speciated AVOCs were varied in the model rather than total DPEC VOCs, reductions in $O_3$ production rate were observed. This highlighted the need to consider the detailed individual VOC speciation when estimating in situ $O_3$ production rate effects. $O_3$ production rate was found to be most sensitive to the OLE2 VOC group, which includes reactive $C_4$-$C_5$ alkenes such as the

but-2-enes and pent-2-ene. This sub-group is forecast to be reduced considerably by 2025 (*ca.* 95%) under the *Ambitious Pollution* scenarios and is likely to strongly influence the reductions in in situ $O_3$ production observed. Between 2030 – 2045, the most ambitious scenario, *Ambitious Pollution 1.5D goals*, did not lead to the largest reductions in $O_3$ production rate. This can be attributed reductions in less reactive species that are present in large amounts in Beijing, such as methanol and the smaller chain alkanes (ALK1 and ALK2). Aerosol surface area (ASA) was found to have a minimal effect on $O_3$ production

rates, with a 69% decrease in ASA leading to a change in $O_3$ production rate of < 1%. $O_3$ production was considerably impacted by possible climate-induced changes in BVOC compound emissions, which was almost entirely driven by changes in isoprene. Doubling the mixing ratios of isoprene, α-pinene and limonene led to the largest increases in $O_3$ production under the *Baseline* scenario, increasing $O_3$ production by 18% in 2060 compared to $O_3$ production projections using changes to anthropogenic VOCs alone. However, it is important to note that the future scenarios presented here are highly uncertain due to their

socioeconomic and political nature and can only be used as a guide. Although estimates for in situ $O_3$ production have been presented in this study, percentage changes in $O_3$ production cannot be applied to $O_3$ concentrations. This is due to the nature of the chemical modelling used, as only instantaneous $O_3$ production can be reproduced, and does not account for background $O_3$, or $O_3$ transported into and out of the measurement site in Beijing. To fully understand how $O_3$ concentrations may vary in future scenarios, further analysis using regional transport models may be required. However, this study provides important

insights into how the in situ chemical processing leading to additional $O_3$ production and destruction in Beijing may vary into the future and highlights the key need to further understand how resultant concentrations from BVOC emissions are expected to change in future years.

**Data availability**

Data are available at http://catalogue.ceda. ac.uk/uuid/7ed9d8a288814b8b85433b0d3fec0300 (last access: Nov 2022).

**Author contributions**

BSN prepared the manuscript with contributions from all authors, FAS, MS and JRH provided measurements and data processing of pollutants used in this study, JFH, ZL, ARR, ACL, JDL and ZS contributed to scientific discussion.


**Competing interests**

The authors declare that they have no conflict of interest.

**Acknowledgements**

This work was financially supported by the UK's Natural Environment Research Council (NERC) COP-AQ project (grant number 2021GRIP02COP-AQ). The project was undertaken on the Viking Cluster, which is a high-performance computing facility provided by the University of York. The authors are grateful for computational support from the University of York High Performance Computing service, Viking, and the Research Computing team.



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
