# Peer review of "The effect of different Climate and Air Quality policies in China on in situ Ozone production in Beijing"

_EGUsphere, 2023_

## Author Comment (AC1)

**RC1**: 'Comment on egusphere-2023-2910', Anonymous Referee #2, 14 Mar 2024

This article describes a detailed chemical box model incorporating the Master Chemical Mechanism to investigate the impact of combined air quality and carbon neutrality policies on O3 formation in Beijing. The results showed that O3 production was most sensitive to changes in the OLE2 group of VOCs, while reducing less reactive but higher concentration species led to larger reductions in O3 production under all scenarios. In general, the idea of the article is clear and the text is smooth. Some details need to be strengthened to meet the publication requirements. Specific comments are as follows.

We thank the reviewer for reading our manuscript carefully and for their useful comments and suggestions. Point by point responses are given below:

Line 38-46: The introduction of O3 are lengthy and need to be reduced to core point.

The detailed description of the human and crop health implications of ozone pollution has been reduced in the manuscript.

When the sentence is connected to the reference, multiple sentence positions are misplaced, such as 'and lead to malnourished populations. (Ainsworth, 2017; IPCC, 2015; Mills et al., 2018)' should be corrected to 'and lead to malnourished populations (Ainsworth, 2017; IPCC, 2015; Mills et al., 2018).'. Please revise throughout the manuscript.

This has been corrected throughout the manuscript.

Line 64-65: 'be' on ' to be account for ' should be removed.

"be" has been removed.

Line 87-94: The box model proposed to evaluate the effect of different Climate on O3 production is suggested to compare with other models.

I have added a reference to this statement. Although chemical box models have been used extensively to explore photochemistry using field data, it is not common to use them to vary model inputs by future Climate projections.

Line 86: "Where comprehensive VOC datasets are available, observationally constrained models incorporating detailed chemical mechanisms, such as the Master Chemical Mechanism (MCM, mcm.york.ac.uk, Jenkin et al., 2015) can be used to

investigate how changes in VOCs, $NO_x$ and aerosol surface area (ASA) may impact in situ $O_3$ formation (Nelson et al., 2021)."

Line 149-150: There is a lack of complete bracket.

Close bracket has been added.

Line 204-220: How to deal with meteorological parameters when adjusting the observational dataset of pollutants in the model.

Unfortunately, there is insufficient inventory data and large uncertainties associated with how the meteorology might change in the future scenarios. As a result, these are kept at observational levels for all scenarios, and we only focus on how changes in the pollutant concentrations may impact future chemistry leading to ozone formation. I have added some text to make this clear to the reader:

Line 204: "This study focuses on the impacts of emission changes on chemical processing, and it is likely that meteorology will also evolve due to a changing climate. However, here we assume that meteorological parameters remain at observed (APHH 2017) levels in all scenarios."

Line 379-382: The points in 'Methanol emissions are projected to increase compared to 2017 levels under the Ambitious Pollution 1.5D goals until 2040' and 'the observed increase in methanol of ca. 25% up to 2040 could have a more pronounced impact on increasing O3 production than smaller incremental changes in more reactive species such as OLE2' is inconsistent with the varies of MEOH group from Figure 3.

The authors would like to thank the reviewer for spotting this error. It is ethane, not methanol, emissions that are projected to increased compared to 2017 under the Ambitious Pollution 1.5D goals until 2040. This has been corrected in the text.

Lines 376 – 389:

The sensitivities of modelled $O_3$ production to these VOCs can be combined with the trends in projected concentrations presented in Figure 4 to explain why $O_3$ production is not reduced the most under the *Ambitious Pollution 1.5D goals* between 2030 - 2045. Ethane (ALK1) emissions are projected to increase compared to 2017 levels under the *Ambitious Pollution 1.5D goals* until 2040 and are a large proportion of AVOCs by mixing ratio (Figure 3). Although increasing ethane mixing ratios by 5% only leads to an increase in $O_3$ production of 0.02%, the observed increase in ethane of *ca.* 25% up to 2040 could have a more pronounced impact on increasing $O_3$ production than smaller incremental changes in more reactive species such as OLE2, that are present in much smaller concentrations by 2025 (< 0.1 ppbV compared to *ca.* 4 ppbV

of ethane). Similarly, although this study shows that modelled $O_3$ production rate is not very sensitive to changes in the ALK2 group, substantial concentrations of this VOC group are observed. As a result, the overall trend in AVOCs (Table 2, excluding isoprene, α-pinene, and limonene) shows a weaker decline for the *Ambitious Pollution 1.5D* scenario which is reflected in the modelled $O_3$ production trend (Figure 4).

Figure S1 and Figure 5 can not be found from this manuscript.

Figure S1 is located in the supporting information document. This is also stated in the text on line 161. Thank you for highlighting an inconsistency in our figure numbering. Line 389 has been changed from "figure 5" to "figure 4", and then subsequently, figure 6 is now called figure 5, figure 7 is figure 6 etc. throughout the manuscript.

The point that aerosols have a minimal effect on O3 production was mentioned in previous studies, such as Tan et al. (2020, 2022). The conclusions here should be compared with other studies.

Comparisons to these additional studies have also been included in the text and referenced as appropriate.

Line 418: This is also consistent with previous studies, which suggest there is no evidence that aerosol chemistry has significant impact on ozone production in the North China Plain, and that aerosol light extinction may cancel out the impacts of aerosols on ozone production in South China (Tan et al., 2020, Tan et al., 2022).

Line 437: O3 subscript writing is wrong.

Subscripting changed.

Line 466-469: The conclusion of 'Between 2030–2045, the most ambitious scenario, Ambitious Pollution 1.5D goals, did not lead to the largest reductions in O3 production rate. This can be attributed reductions in less reactive species that are present in large amounts in Beijing, such as methanol and the smaller chain alkanes (ALK1 and ALK2)' is right? Please check the conclusion with Line 383-387.

This has also been changed, in line with the changes made in Lines 376 – 389. This now reads as follows:

"This can be attributed reductions in less reactive species that are present in large amounts in Beijing, such as the smaller chain alkanes (ALK1 and ALK2)"

---

## Author Comment (AC2)

**RC2**: 'Comment on egusphere-2023-2910', Anonymous Referee #1, 29 Mar 2024

The article titled "The effect of different Climate and Air Quality policies in China on in situ Ozone production in Beijing" by Beth S. Nelson et al. explores how different air quality (AQ) and carbon neutrality (CN) policies in China affect the production of ozone ($O_3$) in Beijing. It delves into the complex relationship between emissions of volatile organic compounds (VOCs), nitrogen oxides (NOx), particulate matter, and the photochemical production of ozone.

In general, this article is well-motivated and written. The intention to evaluate the various emission change scenarios in air quality perspective is prudent. The change in emission is discussed fairly thoroughly in the discussion. However, the change in oxidation capacity and climatology are not factored in the analysis. For example, it is expected a smaller HONO production in the future particularly from heterogeneous reaction sources. In addition, higher temperature and extreme weather are expected due to climate change. I would recommend adding more quantitative discussion from these factors.

The authors would like to thank the reviewer for their feedback. We agree that although this study provides a valuable insight into how ozone production might be expected to change under changing VOC and NOx mixing ratios, there are many compounds not included due to the currently available projects in the DPEC inventory. This means that it is not possible to include HONO projections due to the uncertainty in how HONO levels are expected to change in the future. We also have not considered "extreme" cases, as this is also beyond the scope of this study. To ensure this is fully acknowledged, we have added additional text to section 3.5 ("Sensitivity of changes in concentration of specific AVOCs and ASA on in situ $O_3$ production"), and in the conclusion section of this study, which comments on the quantitative sensitivity of the model to changes in HONO.

Additional text added to Line 204:

"This study focuses on the impacts of emission changes on chemical processing, and it is likely that meteorology will also evolve due to a changing climate. However, here we assume that meteorological parameters remain at observed (APHH 2017) levels in all scenarios."

Additional text added to Line 391:

"It is worth noting that projections in HONO mixing ratios are not included in the DPEC inventory at the time of this study. As a result, mixing ratios of HONO are kept constant under all model scenarios. How HONO might change under different future scenarios is highly uncertain. Although it is generally expected that HONO mixing ratios will correlate with changing $NO_x$, its formation is dependent on multiple factors including photolysis rates, heterogeneous chemistry on particulate matter and surfaces, and meteorological conditions (Sander and Peterson, 1984; Lee et al., 2016). However, a recent study found that reductions in $NO_x$ during the COVID-19 lockdowns in the Chinese megacity of Zhengzhou did not lead to comparable reductions in HONO during the day (Wang et al., 2024). When HONO is increased by 5% in the model, we observe a 1.9% increase in $O_3$ production. This shows that $O_3$ production is highly sensitive to changes in HONO and emphasises the importance of improving our

understanding of how HONO might be expected to change under different socioeconomic, climate, and carbon neutrality goals."

Additional text added to Line 489:

"The focus of this study is the impacts of the changing VOC emissions scenarios on photochemical $O_3$ formation. However, there are several other important factors that will evolve in a changing climate that will likely affect the formation and concentrations of $O_3$, such as meteorology and extreme temperature and biomass burning events impacting urban areas such as Beijing. Heterogeneous sources of HONO, an important source of OH radical in urban environments (Lee et al., 2016) are also likely to change, impacting on urban oxidising capacity and hence $O_3$ formation. However, how these factors are likely to change are highly uncertain, and should be looked at further in future studies."